# Design and Verification of a New Universal Active Filter Based on the Current Feedback Operational Amplifier and Commercial AD844 Integrated Circuit

**DOI:** 10.3390/s23198258

**Published:** 2023-10-05

**Authors:** Hua-Pin Chen, I-Chyn Wey, Liang-Yen Chen, Cheng-Yueh Wu, San-Fu Wang

**Affiliations:** 1Department of Electronic Engineering, Ming Chi University of Technology, New Taipei 24301, Taiwan; m12158006@o365.mcut.edu.tw; 2Department of Electrical Engineering, Chang Gung University, Taoyuan City 33302, Taiwan; icwey@mail.cgu.edu.tw (I.-C.W.); b1121255@cgu.edu.tw (L.-Y.C.); 3Department of Electronic Engineering, National Chin-Yi University of Technology, Taichung 41170, Taiwan; sf_wang@ncut.edu.tw

**Keywords:** analog circuit design, chip implementation, voltage-mode filter, universal filter

## Abstract

This paper presents a triple-input and four-output type voltage-mode universal active filter based on three current-feedback operational amplifiers (CFOAs). The filter employs three CFOAs, two grounded capacitors, and six resistors. The filter structure has three high-input and three low-output impedances that simultaneously provide band-reject, high-pass, low-pass, and band-pass filtering functions with single-input and four-output type and also implements an all-pass filtering function by connecting three input signals to one input without the use of voltage inverters or switches. The same circuit configuration enables two unique filtering functions: low-pass notch and high-pass notch. Three CFOAs with three high-input and low-output impedance terminals enable cascading without voltage buffers. The circuit is implemented using three commercial off-the-shelf AD844 integrated circuits, two grounded capacitors, and six resistors and further implemented as a CFOA-based chip using three CFOAs, two grounded capacitors, and six resistors. The CFOA-based chip has lower power consumption and higher integration than the AD844-based filter. The circuit was simulated using OrCAD PSpice to verify the AD844-based filter and Synopsys HSpice for post-layout simulation of the CFOA-based chip. The theoretical analysis is validated and confirmed by measurements on an AD844-based filter and a CFOA-based chip.

## 1. Introduction

In electrical circuits and systems, filters play an essential role in sensors and signal processing and have received considerable attention in recent years because signals acquired through sensing elements must be filtered out of external noise using filters [1,2,3,4,5,6]. The technical literature [7] describes a conceptual scheme for sensing applications in phase-sensitive detection technology, where two low-pass (LP) filters are used to select the frequency range and eliminate out-of-band noise from the sensor device signal. For sensors, instrumentation and measurement systems, and electrical systems, filters reduce external noise, eliminate interference, improve signal quality, and maintain signal integrity [8,9,10].

Active circuits with high-performance characteristics are of great interest [11,12,13,14,15,16,17,18,19,20,21,22,23,24,25,26], especially the AD844AN integrated circuit (IC) using CFOA, which is beneficial to rapid verification of the designed circuits [27,28,29,30,31]. Typically, the voltage-mode universal second-order filters can realize all-pass (AP), band-reject (BR), high-pass (HP), LP, and band-pass (BP) filtering functions by properly selecting different input signals. In contrast, the voltage-mode multifunction second-order filters can simultaneously realize HP, LP, and BP filtering functions [32]. Two unique filtering functions, low-pass notch (LPN) and high-pass notch (HPN), are particular BR filtering functions. The universal active filter (UAF) includes seven different filtering functions, LP, BP, HP, BR, LPN, HPN, and AP, whose versatility and practicality make them attractive to designers of active filters. A prominent application of CFOA-based filters concerns the voltage-mode UAF configuration that provides seven filtering capabilities. The UAF configuration features seven different filtering functions that eliminate external noise, reduce interference, improve signal quality, and ensure signal integrity. Many researchers use commercial off-the-shelf AD844AN ICs to verify the accuracy performance of electronic circuit designs because they are high quality, inexpensive, and readily available [32,33,34,35,36,37,38,39]. The CFOA contains the well-known active element second-generation current conveyor (CCII) in the input stage and a voltage buffer in the output stage, solving the low output impedance issue of the CCII [40,41,42,43]. Alternatively, CFOA can be implemented on a commercially available off-the-shelf AD844AN IC to facilitate circuit design verification. Up to now, researchers have mainly focused on implementing voltage-mode second-order filters employing CFOAs. With attention to the voltage-mode universal/multifunction CFOA-based filters [32,33,34,35,36,37,38,39,40,41,42,43], researchers are also constantly developing new CFOA-based filters to increase their convenience, versatility, and universality. In [32], the multifunction filter based on four CFOAs simultaneously realizes voltage-mode LP, BP, and HP filtering functions. The topology cannot implement the BR and AP filtering functions from the same configuration. In [33], the voltage-mode universal filter based on three CFOAs is proposed. The proposed topology uses two floating capacitors. The topology in [34] can simultaneously realize BR, LP, BP, and HP filtering functions with high input impedance. The proposed topology uses four CFOAs, two grounded capacitors, and four resistors. Another high input impedance multifunction filter employing three CFOAs and six passive elements was proposed in [35]. The proposed topology has the disadvantage that a capacitor is a direct connection to the X terminal of a CFOA, thus limiting the frequency of operation of the circuit [16]. In [36], a voltage-mode UAF with high input impedance based on four CFOAs was proposed. This topology requires four CFOA active elements and a BR/AP filtering function switch. In [37,38,39,40,41,42], some voltage-mode multifunction filters based on three CFOAs simultaneously realize LP, BP, and HP/BR filtering functions. These topologies still cannot implement the AP filtering function from the same configuration. In 2022, three voltage-mode UAFs based on four CFOAs were proposed in the technical literature [43]. These three CFOA-based voltage-mode UAF circuits use four CFOAs, two grounded capacitors, five/six resistors, and one/two switches. Each design topology requires three CFOA active elements to simultaneously implement LP, BP, BR, and one CFOA with one or two HP/AP filter function switches. After carefully studying the literature [32,33,34,35,36,37,38,39,40,41,42,43] based on single-input or multiple-input voltage-mode second-order filters, LP, BP, HP, BR, LPN, HPN, and AP filtering functions implemented from the same configuration were not studied. The practical analog filter design must study the convenience, versatility, and modularity of voltage-mode UAF.

This paper proposes a voltage-mode UAF with three high-input and low-output impedances based on three CFOAs, two grounded capacitors, and six resistors. The proposed CFOA-based voltage-mode UAF realizes LP, BP, HP, BR, LPN, HPN, and AP filtering functions from the same configuration and offers the availability of input voltages at high input impedance terminals. Furthermore, the proposed voltage-mode UAF was implemented into a single CFOA-based chip. The CFOA-based voltage-mode UAF chip has lower power consumption and higher integration than the AD844-based filter. Contrary to the previously reported voltage-mode universal/multifunction second-order filters [32,33,34,35,36,37,38,39,40,41,42,43], the proposed voltage-mode UAF meets the following advantages.

(i)The circuit uses only three CFOAs, two grounded capacitors, and six resistors with no switches.(ii)Utilizing only two grounded capacitors makes the circuit suitable for IC implementation.(iii)Use the same circuit configuration to implement voltage-mode second-order LP, BP, HP, BR, LPN, HPN, and AP filtering functions.(iv)Three high-input and low-output impedances are suitable for cascading voltage-mode operation capability without voltage buffers.(v)Simultaneously realize the voltage-mode second-order LP, BP, HP, and BR filtering functions without requiring component matching conditions.(vi)No capacitance is connected in series at terminal X of the CFOA.(vii)No voltage inverter is required for the AP filtering function.(viii)The filter parameters of resonance angular frequency (ω_o_) and quality factor (Q) independently control the particular case.(ix)The circuit has low active and passive sensitivity performance.(x)Integrate the voltage-mode UAF into a single CFOA-based chip.

Table 1 compares the proposed voltage-mode UAF and the previous CFOA-based biquadratic filters [32,33,34,35,36,37,38,39,40,41,42,43]. The proposed CFOA-based voltage-mode UAF simultaneously satisfies all of the tenth properties. In addition, the circuit is further implemented in a single CFOA-based voltage-mode UAF chip with lower power consumption and higher integration than the AD844-based filter. Compared to CFOA-based voltage-mode filters in the recent technical literature [39,40,41,42,43], the proposed CFOA-based voltage-mode UAF does not require any switch and achieves LP, BP, HP, BR, LPN, HPN, and AP filtering functions from the same configuration. Moreover, the circuit simultaneously realizes LP, NP, HP, and BR filtering functions. The CFOA-based voltage-mode UAF chip is manufactured with TSMC 180 nm 1P6M CMOS process technology, featuring low power consumption and high integration. Measurements using three commercial off-the-shelf AD844AN ICs and a CFOA-based voltage-mode UAF chip validate and confirm the theoretical analysis.

## 2. Circuit Descriptions and Realizations

The ideal CFOA is a four-port active component with two input ports, *X* and *Y*, and two output ports, *Z* and *W*, as shown in Figure 1. The port relations of CFOA can be characterized by V_X_ = V_Y_, V_W_ = V_Z_, I_Y_ = 0, and I_X_ = I_Z_ [43]. The ideal CFOA is logically representable as a voltage buffer between the Y and X terminals, a current follower between the X and Z terminals, and a subsequent voltage buffer between the Z and W ports. Moreover, CFOA can be implemented in hardware using commercially available off-the-shelf AD844AN IC [44].

### 2.1. Proposed Voltage-Mode UAF Configuration

The proposed voltage-mode UAF configuration is shown in Figure 2, which consists of three CFOAs, two grounded capacitors, and six resistors. In Figure 2, the input signals V_i1_, V_i2_, and V_i3_ are connected to the high input terminal of each Y terminal of the three CFOAs. The output voltages V_o1_, V_o2_, and V_o3_ are connected to the low output terminal of each W terminal of the three CFOAs. High-input and low-output impedances can be used for cascaded voltage-mode operation without requiring any voltage buffers. Routine analysis of the voltage-mode UAF in Figure 2 gives the following four output voltages:
(1)$$V_{o1}=\frac{s\frac{1}{C_{1}R_{3}}V_{i1}+\frac{R_{1}}{C_{1}C_{2}R_{2}R_{3}R_{4}}V_{i2}-s\frac{R_{1}}{C_{1}R_{2}R_{3}}V_{i3}}{s^{2}+s\frac{1}{C_{1}R_{3}}+\frac{R_{1}}{C_{1}C_{2}R_{2}R_{3}R_{4}}}$$
(2)$$V_{o2}=\frac{-\frac{1}{C_{1}C_{2}R_{3}R_{4}}V_{i1}+(s\frac{1}{C_{2}R_{4}}+\frac{1}{C_{1}C_{2}R_{3}R_{4}})V_{i2}+\frac{R_{1}}{C_{1}C_{2}R_{2}R_{3}R_{4}}V_{i3}}{s^{2}+s\frac{1}{C_{1}R_{3}}+\frac{R_{1}}{C_{1}C_{2}R_{2}R_{3}R_{4}}}$$
(3)$$V_{o3}=\frac{(s\frac{1}{C_{1}R_{3}}+\frac{R_{1}}{C_{1}C_{2}R_{2}R_{3}R_{4}}{)V}_{i1}-s\frac{R_{1}}{C_{2}R_{2}R_{4}}V_{i2}+s^{2}\frac{R_{1}}{R_{2}}V_{i3}}{s^{2}+s\frac{1}{C_{1}R_{3}}+\frac{R_{1}}{C_{1}C_{2}R_{2}R_{3}R_{4}}}$$
(4)$$V_{o4}=\frac{1}{R_{5}+R_{6}}(R_{6}V_{o2}+R_{5}V_{o3})$$

According to (1)–(4), when the two input voltages of V_i1_ and V_i2_ are grounded, and V_i3_ is provided as the input signal of V_in_, four different filtering functions can be realized simultaneously.
(5)$$\frac{V_{o1}}{V_{in}}=a_{1}\frac{-s\frac{\omega_{o}}{Q}}{s^{2}+s\frac{\omega_{o}}{Q}+\omega_{o}^{2}}=\frac{R_{1}}{R_{2}}(\frac{-s\frac{1}{C_{1}R_{3}}}{s^{2}+s\frac{1}{C_{1}R_{3}}+\frac{R_{1}}{C_{1}C_{2}R_{2}R_{3}R_{4}}})$$
(6)$$\frac{V_{o2}}{V_{in}}=\frac{\omega_{o}^{2}}{s^{2}+s\frac{\omega_{o}}{Q}+\omega_{o}^{2}}=\frac{\frac{R_{1}}{C_{1}C_{2}R_{2}R_{3}R_{4}}}{s^{2}+s\frac{1}{C_{1}R_{3}}+\frac{R_{1}}{C_{1}C_{2}R_{2}R_{3}R_{4}}}$$
(7)$$\frac{V_{o3}}{V_{in}}=a_{1}\frac{s^{2}}{s^{2}+s\frac{\omega_{o}}{Q}+\omega_{o}^{2}}=\frac{R_{1}}{R_{2}}(\frac{s^{2}}{s^{2}+s\frac{1}{C_{1}R_{3}}+\frac{R_{1}}{C_{1}C_{2}R_{2}R_{3}R_{4}}})$$
(8)$$\frac{V_{o4}}{V_{in}}=a_{2}\frac{s^{2}+\omega_{z}^{2}}{s^{2}+s\frac{\omega_{o}}{Q}+\omega_{o}^{2}}=\frac{R_{1}R_{5}}{R_{2}(R_{5}+R_{6})}(\frac{s^{2}+\frac{R_{6}}{C_{1}C_{2}R_{3}R_{4}R_{5}}}{s^{2}+s\frac{1}{C_{1}R_{3}}+\frac{R_{1}}{C_{1}C_{2}R_{2}R_{3}R_{4}}})$$

From (5) to (8), the filter parameters bandwidth (BW), ω_o_, Q, passband gain a_1_ and a_2_, and BR frequency ω_z_ are given by
(9)$$\mathrm{BW}=\frac{1}{C_{1}R_{3}},\;\omega_{o}=\sqrt{\frac{R_{1}}{C_{1}C_{2}R_{2}R_{3}R_{4}}},\;Q=\sqrt{\frac{C_{1}R_{1}R_{3}}{C_{2}R_{2}R_{4}}}$$
(10)$$a_{1}=\frac{R_{1}}{R_{2}},\;a_{2}=\frac{R_{1}R_{5}}{R_{2}(R_{5}+R_{6})}$$
(11)$$\omega_{z}=\sqrt{\frac{R_{6}}{C_{1}C_{2}R_{3}R_{4}R_{5}}}$$

Based on (5) to (7), an inverting band-pass (IBP) filtering function with R_1_/R_2_ passband gain is obtained at V_o1_, a non-inverting LP (NLP) filtering function with unity passband gain is obtained at V_o2_, and a non-inverting HP (NHP) filtering function with R_1_/R_2_ passband gain is obtained at V_o3_. According to (8), the following three BR filtering functions are obtained.

(a)If R_1_ = R_2_ and R_5_ = R_6_, the regular BR filtering function with 1/2 passband gain can be realized in (12).



(12)
$$\frac{V_{o4}}{V_{in}}=\frac{1}{2}(\frac{s^{2}+\frac{1}{C_{1}C_{2}R_{3}R_{4}}}{s^{2}+s\frac{1}{C_{1}R_{3}}+\frac{1}{C_{1}C_{2}R_{3}R_{4}}})$$



(b)If R_5_ < R_6_ and ω_z_ > ω_o_, the LPN filtering function can be obtained in (8).(c)If R_5_ > R_6_ and ω_z_ < ω_o_, the HPN filtering function can also be obtained in (8).

By connecting three input signals of V_i1_, V_i2_, and V_i3_ into one input signal and selecting the matching element condition of R_1_ = R_2_ and C_1_R_3_ = 2C_2_R_4_, the voltage-mode UAF of V_o3_ performs the following non-inverting AP (NAP) filtering function.
(13)$$V_{o3}=\frac{s^{2}-s\frac{1}{C_{1}R_{3}}+\frac{1}{C_{1}C_{2}R_{3}R_{4}}}{s^{2}+s\frac{1}{C_{1}R_{3}}+\frac{1}{C_{1}C_{2}R_{3}R_{4}}}$$

According to (9), the voltage-mode UAF parameters BW, ω_o,_ and Q can be controlled orthogonally by R_1_ = R_3_ = R_a_. In this particular case, (9) becomes
(14)$$\mathrm{BW}=\frac{1}{C_{1}R_{a}},\;\omega_{o}=\sqrt{\frac{1}{C_{1}C_{2}R_{2}R_{4}}},\;Q=R_{a}\sqrt{\frac{C_{1}}{C_{2}R_{2}R_{4}}}$$

Equation (14) expresses that R_a_ independently controls the parameters BW and Q without affecting the parameter ω_o_. When R_3_ = R_4_ = R_b_, the parameters ω_o_ and Q in (9) become
(15)$$\omega_{o}=\frac{1}{R_{b}}\sqrt{\frac{R_{1}}{C_{1}C_{2}R_{2}}},\;Q=\sqrt{\frac{C_{1}R_{1}}{C_{2}R_{2}}}$$

Equation (15) expresses that R_b_ independently controls the parameter ω_o_ without affecting the parameter Q. Equations (14) and (15) show that the voltage-mode UAF parameters Q and ω_o_ are independently controlled by R_a_ and R_b_, respectively.

### 2.2. Effects of CFOA Non-Idealities on Voltage-Mode UAF Characteristics

Assuming the non-ideal CFOA terminal characteristics as V_X_ = βV_Y_, V_W_ = γV_Z_, and I_X_ = αI_Z_, where β = 1 − ε_βv_, γ = 1 − ε_γv_ and α = 1 − ε_αi_. Here, ε_βv_ (|ε_βv_| << 1) and ε_γv_ (|ε_γv_| << 1) denote the voltage tracking errors of CFOA, and ε_αi_ (|ε_αi_| << 1) represent the current tracking error of non-ideal CFOA [31]. Considering the non-ideal CFOA terminal characteristics in Figure 2, the analysis of voltage-mode UAF yields the denominator of the non-ideal voltage transfer function as follows.
(16)$$D(s)=s^{2}+s\frac{\alpha_{1}\gamma_{1}\gamma_{3}}{C_{1}R_{3}}+\frac{\alpha_{1}\alpha_{2}\alpha_{3}\gamma_{1}\gamma_{2}\gamma_{3}R_{1}}{C_{1}C_{2}R_{2}R_{3}R_{4}}$$

The voltage-mode UAF parameters ω_o_ and Q, in the case of non-ideal CFOA terminal characteristics, are
(17)$$\omega_{o}=\sqrt{\frac{\alpha_{1}\alpha_{2}\alpha_{3}\gamma_{1}\gamma_{2}\gamma_{3}R_{1}}{C_{1}C_{2}R_{2}R_{3}R_{4}}},\;Q=\sqrt{\frac{\alpha_{2}\alpha_{3}\gamma_{2}C_{1}R_{1}R_{3}}{\alpha_{1}\gamma_{1}\gamma_{3}C_{2}R_{2}R_{4}}}$$

According to the definition of [43], the active and passive sensitivity parameters ω_o_ and Q of voltage-mode UAF are calculated as follows.
(18)$$S_{\alpha_{1}}^{\omega_{o}}=S_{\alpha_{2}}^{\omega_{o}}=S_{\alpha_{3}}^{\omega_{o}}=S_{\gamma_{1}}^{\omega_{o}}=S_{\gamma_{2}}^{\omega_{o}}=S_{\gamma_{3}}^{\omega_{o}}=S_{R_{1}}^{\omega_{o}}=0.5$$
(19)$$S_{C_{1}}^{\omega_{o}}=S_{C_{2}}^{\omega_{o}}=S_{R_{2}}^{\omega_{o}}=S_{R_{3}}^{\omega_{o}}=S_{R_{4}}^{\omega_{o}}=-0.5$$
(20)$$S_{\alpha_{2}}^{Q}=S_{\alpha_{3}}^{Q}=S_{\gamma_{2}}^{Q}=S_{C_{1}}^{Q}=S_{R_{1}}^{Q}=S_{R_{3}}^{Q}=0.5$$
(21)$$S_{\alpha_{1}}^{Q}=S_{\gamma_{1}}^{Q}=S_{\gamma_{3}}^{Q}=S_{C_{2}}^{Q}=S_{R_{2}}^{Q}=S_{R_{4}}^{Q}=-0.5$$

From the results, the voltage-mode UAF exhibits low active and passive sensitivities.

If V_i1_ and V_i2_ are grounded, and only V_i3_ is provided as the input signal of V_in_, the four different filtering functions for non-ideal voltage and current gains become
(22)$$\frac{V_{o1}}{V_{in}}=\frac{R_{1}}{R_{2}}(\frac{-s\frac{\alpha_{1}\alpha_{3}\beta_{3}\gamma_{3}}{C_{1}R_{3}}}{s^{2}+s\frac{\alpha_{1}\gamma_{1}\gamma_{3}}{C_{1}R_{3}}+\frac{\alpha_{1}\alpha_{2}\alpha_{3}\gamma_{1}\gamma_{2}\gamma_{3}R_{1}}{C_{1}C_{2}R_{2}R_{3}R_{4}}})$$
(23)$$\frac{V_{o2}}{V_{in}}=\frac{\frac{\alpha_{1}\alpha_{2}\alpha_{3}\beta_{1}\gamma_{1}\gamma_{3}R_{1}}{C_{1}C_{2}R_{2}R_{3}R_{4}}}{s^{2}+s\frac{\alpha_{1}\gamma_{1}\gamma_{3}}{C_{1}R_{3}}+\frac{\alpha_{1}\alpha_{2}\alpha_{3}\gamma_{1}\gamma_{2}\gamma_{3}R_{1}}{C_{1}C_{2}R_{2}R_{3}R_{4}}}$$
(24)$$\frac{V_{o3}}{V_{in}}=\frac{R_{1}}{R_{2}}(\frac{\alpha_{3}\beta_{3}s^{2}}{s^{2}+s\frac{\alpha_{1}\gamma_{1}\gamma_{3}}{C_{1}R_{3}}+\frac{\alpha_{1}\alpha_{2}\alpha_{3}\gamma_{1}\gamma_{2}\gamma_{3}R_{1}}{C_{1}C_{2}R_{2}R_{3}R_{4}}})$$
(25)$$\frac{V_{o4}}{V_{in}}=\frac{R_{1}R_{5}}{R_{2}(R_{5}+R_{6})}(\frac{\alpha_{3}\beta_{3}s^{2}+\frac{\alpha_{1}\alpha_{2}\alpha_{3}\beta_{1}\gamma_{1}\gamma_{3}R_{6}}{C_{1}C_{2}R_{3}R_{4}R_{5}}}{s^{2}+s\frac{\alpha_{1}\gamma_{1}\gamma_{3}}{C_{1}R_{3}}+\frac{\alpha_{1}\alpha_{2}\alpha_{3}\gamma_{1}\gamma_{2}\gamma_{3}R_{1}}{C_{1}C_{2}R_{2}R_{3}R_{4}}})$$

The CFOA terminal characteristics generally have various parasitic impedances [43]. The parasitic impedances of these non-ideal CFOAs can affect the performance of the proposed voltage-mode UAF. Figure 3 illustrates the non-ideal CFOA model for analyzing the parasitic impedances and their effect on the voltage-mode UAF configuration. Using the CFOA non-ideal model, the parasitic impedance effect of the proposed voltage-mode UAF is analyzed, as shown in Figure 4. Reanalyzing the voltage-mode UAF in Figure 4, the following non-ideal four output node voltages are obtained:
(26)$$[s(C_{1}+C_{Z1})+\frac{1}{R_{Z1}}]V_{o1}+\frac{V_{o3}}{R_{3}+R_{X1}}=\frac{V_{i1}}{R_{3}+R_{X1}}$$
(27)$$(\frac{1}{R_{4}+R_{X2}})V_{o1}+[s(C_{2}+C_{Z2})+\frac{1}{R_{Z2}}]V_{o2}=\frac{V_{i2}}{R_{4}+R_{X2}}$$
(28)$$-\frac{1}{R_{1}}V_{o1}+(\frac{1}{R_{2}+R_{X3}})V_{o2}+(sC_{Z3}+\frac{1}{R_{Z3}}+\frac{1}{R_{1}})V_{o3}=\frac{V_{i3}}{R_{2}+R_{X3}}$$
(29)$$V_{o4}=\frac{1}{R_{5}+R_{6}}(R_{6}V_{o2}+R_{5}V_{o3})$$

Equations (26)–(29) show that several parasitic resistances of R_X1_, R_X2_, R_X3_, R_Z1_, R_Z2_, and R_Z3_ and three parasitic capacitances of C_Z1_, C_Z2_, and C_Z3_ will affect the voltage-mode UAF. In Figure 4, the proposed voltage-mode UAF has the attractive advantage that capacitors C_1_ and C_2_ are grounded, and resistors R_2_, R_3_, and R_4_ are connected to the X-terminal of the CFOA, respectively. The main advantage of the proposed voltage-mode UAF topology is that the two parasitic capacitance effects of C_Z1_ and C_Z2_ can be absorbed by the two grounded capacitors of C_1_ and C_2_, and the three parasitic resistances of R_X1_, R_X2_, and R_X3_ can also be absorbed by three series resistors of R_3_, R_4_, and R_2_, respectively. It is worth noting that the parasitic capacitance C_Z3_ and the parasitic resistances R_Z1_, R_Z2_, and R_Z3_ affect the operating frequency range of voltage-mode UAF. If the conditions of s(C_1_ + C_Z1_) >> 1/R_Z1_, s(C_2_ + C_Z2_) >> 1/R_Z2_, and sC_Z3_(R_1_//R_Z3_) << 1 are satisfied, the influence of the parasitic resistances and capacitances on voltage-mode UAF topology can be reduced. Therefore, the valid operating frequency range of the voltage-mode UAF needs to be considered as follows.
(30)$$10\times \max\left\{\frac{1}{2\pi (C_{1\;}+C_{Z1})R_{Z1}},\;\frac{1}{2\pi (C_{2\;}+C_{Z2})R_{Z2}}\right\}\;\leq \;f\;\leq \;\frac{0.1}{2\pi C_{Z3}(R_{1}//R_{Z3})}$$

According to (30), the non-ideal three output node voltages of (26) to (28) are simplified as
(31)$${s(C}_{1}{+C}_{Z1}{)V}_{o1}+\frac{V_{o3}}{R_{3}{+R}_{X1}}=\frac{V_{i1}}{R_{3}{+R}_{X1}}$$
(32)$$(\frac{1}{R_{4}{+R}_{X2}}{)V}_{o1}{+s(C}_{2}{+C}_{Z2}{)V}_{o2}=\frac{V_{i2}}{R_{4}{+R}_{X2}}$$
(33)$$\frac{1}{R_{1}}V_{o1}+(\frac{1}{R_{2}{+R}_{X3}}{)V}_{o2}+(\frac{1}{R_{Z3}}+\frac{1}{R_{1}}{)V}_{o3}=\frac{V_{i3}}{R_{2}{+R}_{X3}}$$

In this case, the three non-ideal voltage transfer functions, the denominator D_n_(s), and the non-ideal filter parameters of ω_on_ and Q_n_ were obtained as
(34)$$\begin{array}{l} V_{o1}=\frac{1}{D_{n}\left(s\right)}[s\frac{1}{{(C}_{1}+C_{Z1}{)(R}_{3}+R_{X1})}V_{i1}\\ \;+\frac{R_{1}{//R}_{Z3}}{{(C}_{1}+C_{Z1}{)(C}_{2}+C_{Z2}{)(R}_{2}+R_{X3}{)(R}_{3}+R_{X1}{)(R}_{4}+R_{X2})}V_{i2}\\ \;-s\frac{R_{1}{//R}_{Z3}}{{(C}_{1}+C_{Z1}{)(R}_{2}+R_{X3}{)(R}_{3}+R_{X1})}V_{i3}] \end{array}$$
(35)$$\begin{array}{l} V_{o2}=\frac{1}{D_{n}\left(s\right)}[-\frac{1}{{(C}_{1}+C_{Z1}{)(C}_{2}+C_{Z2}{)(R}_{3}+R_{X1}{)(R}_{4}+R_{X2})}V_{i1}\\ \;+s\frac{1}{{(C}_{2}+C_{Z2}{)(R}_{4}+R_{X2})}V_{i2}\\ \;+\frac{R_{1}{//R}_{Z3}}{{(C}_{1}+C_{Z1}{)(C}_{2}+C_{Z2}{)(R}_{3}+R_{X1}{)(R}_{4}+R_{X2}{)R}_{1}}V_{i2}\\ \;+\frac{R_{1}{//R}_{Z3}}{{(C}_{1}+C_{Z1}{)(C}_{2}+C_{Z2}{)(R}_{2}+R_{X3}{)(R}_{3}+R_{X1}{)(R}_{4}+R_{X2})}V_{i3}] \end{array}$$
(36)$$\begin{array}{l} V_{o3}=\frac{1}{D_{n}\left(s\right)}[s\frac{R_{1}{//R}_{Z3}}{{(C}_{1}+C_{Z1}{)(R}_{3}+R_{X1}{)R}_{1}}V_{i1}\\ \;+\frac{R_{1}{//R}_{Z3}}{{(C}_{1}+C_{Z1}{)(C}_{2}+C_{Z2}{)(R}_{2}+R_{X3}{)(R}_{3}+R_{X1}{)(R}_{4}+R_{X2})}V_{i1}\\ \;-s\frac{R_{1}{//R}_{Z3}}{{(C}_{2}+C_{Z2}{)(R}_{2}+R_{X3}{)(R}_{4}+R_{X2})}V_{i2}\\ \;+s^{2}\frac{R_{1}{//R}_{Z3}}{R_{2}+R_{X3}}V_{i3}] \end{array}$$
(37)$$D_{n}{(s)=s}^{2}+s\frac{R_{1}{//R}_{Z3}}{{(C}_{1}{+C}_{Z1}{)(R}_{3}{+R}_{X1}{)R}_{1}}+\frac{R_{1}{//R}_{Z3}}{{(C}_{1}{+C}_{Z1}{)(C}_{2}{+C}_{Z2}{)(R}_{2}{+R}_{X3}{)(R}_{3}{+R}_{X1}{)(R}_{4}{+R}_{X2})}$$
(38)$$\omega_{\mathrm{on}}=\sqrt{\frac{R_{1}{//R}_{Z3}}{{(C}_{1}{+C}_{Z1}{)(C}_{2}{+C}_{Z2}{)(R}_{2}{+R}_{X3}{)(R}_{3}{+R}_{X1}{)(R}_{4}{+R}_{X2})}}$$
(39)$$Q_{n}{=R}_{1}\sqrt{\frac{{(C}_{1}{+C}_{Z1}{)(R}_{3}{+R}_{X1})}{{(C}_{2}{+C}_{Z2}{)(R}_{1}{//R}_{Z3}{)(R}_{2}{+R}_{X3}{)(R}_{4}{+R}_{X2})}}$$

If V_i1_ and V_i2_ are grounded, and only V_i3_ is provided as the input signal of V_in_, the four different filtering functions for non-ideal voltage outputs become
(40)$$\frac{V_{o1}}{V_{\mathrm{in}}}=\frac{R_{1}{//R}_{Z3}}{R_{2}{+R}_{X3}}(\frac{-s\frac{1}{{(C}_{1}{+C}_{Z1}{)(R}_{3}{+R}_{X1})}}{D_{n}})$$
(41)$$\frac{V_{o2}}{V_{\mathrm{in}}}=\frac{\frac{R_{1}{//R}_{Z3}}{{(C}_{1}{+C}_{Z1}{)(C}_{2}{+C}_{Z2}{)(R}_{2}{+R}_{X3}{)(R}_{3}{+R}_{X1}{)(R}_{4}{+R}_{X2})}}{D_{n}}$$
(42)$$\frac{V_{o3}}{V_{\mathrm{in}}}=\frac{R_{1}{//R}_{Z3}}{R_{2}{+R}_{X3}}(\frac{s^{2}}{D_{n}})$$
(43)$$\frac{V_{o4}}{V_{\mathrm{in}}}=\frac{{(R}_{1}{//R}_{Z3}{)R}_{5}}{{(R}_{2}{+R}_{X3}{)(R}_{5}{+R}_{6})}(\frac{s^{2}+\frac{R_{6}}{{(C}_{1}{+C}_{Z1}{)(C}_{2}{+C}_{Z2}{)(R}_{3}{+R}_{X1}{)(R}_{4}{+R}_{X2}{)R}_{5}}}{D_{n}})$$

As shown in (30), the voltage-mode UAF must operate within the effective operating frequency range to minimize the effects of the non-ideal CFOA parasitic resistances and capacitances.

## 3. Simulation and Experimental Results

The proposed voltage-mode UAF efficiency and flexibility were demonstrated using commercially available off-the-shelf AD844AN ICs and on-chip design measurements to validate the theoretical analysis. The AD844-based voltage-mode UAF has a supply voltage of 12 V (±6 V) and a power consumption of 168 mW. The on-chip CMOS CFOA-based VM-UAF has a supply voltage of 1.8 V (±0.9 V) and a power consumption of 3.6 mW. The performance of the AD844-based voltage-mode UAF and on-chip CMOS CFOA-based voltage-mode UAF was evaluated using the OrCAD PSpice software and the Synopsys HSpice simulation design environment using TSMC 180 nm 1P6M CMOS technology, respectively. Simulations were performed using the built-in library of AD844/AD model parameters of OrCAD PSpice software, and OrCAD PSpice software features sensitivity/Monte Carlo analysis capabilities. The experimental setups of AD844-based voltage-mode UAF and on-chip CMOS CFOA-based voltage-mode UAF are shown in Figure 5 and Figure 6, respectively. Frequency domain simulations evaluate the AD844-based voltage-mode UAF and on-chip CMOS CFOA-based voltage-mode UAF. Measurements on an AD844-based voltage-mode UAF and a CFOA-based voltage-mode UAF chip validate and confirm the theoretical analysis.

### 3.1. The AD844-Based Voltage-Mode UAF Simulation and Measurement Results

The passive components of the AD844-based voltage-mode UAF are selected as C_1_ = C_2_ = 100 pF and R_i_ = 10 kΩ (i = 1 to 6) with a resonance frequency of f_o_ = 159.15 kHz. Regarding the AD844 datasheet [44], the X terminal parasitic resistance of AD844 is R_X_ = 50 Ω. The Z terminal parasitic resistance and parasitic capacitance of AD844 are R_Z_ = 3 MΩ and C_Z_ = 4.5 pF, respectively. According to (30), the effective operating frequency range of the AD844-based voltage-mode UAF is 1.74 kHz to 885.37 kHz. Figure 7 shows the simulated frequency spectrum of the IBP filtering response at V_o1_. As shown in Figure 7, the total harmonic distortion (THD) is calculated as 0.6% for a sinusoidal input voltage of 2.4 V_pp_. Figure 8 shows the measured frequency spectrum of the IBP filtering response at V_o1_. As shown in Figure 8, the THD is calculated as 1.68% for a sinusoidal input voltage of 5.2 V_pp,_ and the measured spurious-free dynamic range is 37.62 dBc. The gain and phase of AD844-based VM-UAF in the frequency domain simulations are shown in Figure 9, Figure 10, Figure 11 and Figure 12. Figure 13, Figure 14, Figure 15 and Figure 16 also show the measurements for the AD844-based voltage-mode UAF. The experimental and simulation results of the AD844-based voltage-mode UAF relative to the theoretical analysis are shown in Figure 17, Figure 18, Figure 19 and Figure 20. It can be seen that the experimental and simulation results of the AD844-based voltage-mode UAF are close to the theoretical predictions.

According to (14), R_1_ = R_3_ = R_a_ can independently control the Q value in the AD844-based voltage-mode UAF without affecting the parameter ω_o_. Thus, with fixed C_1_ = C_2_ = 300 pF and R_2_ = R_4_ = 4 kΩ, the required R_a_ values are 3.2 kΩ, 5.44 kΩ, 7.68 kΩ, and 10 kΩ for selected Q values of 0.8, 1.36, 1.92, and 2.5, respectively. Figure 21, Figure 22 and Figure 23 show the behavior of the AD844-based VM-UAF quality factor independently controlled by R_a_ when V_i3_ = V_in_ and V_i1_ = V_i2_ = 0. According to (15), R_3_ = R_4_ = R_b_ can independently control the resonance frequency value in the AD844-based voltage-mode UAF without affecting the parameter Q. Thus, with fixed C_1_ = C_2_ = 100 pF and R_1_ = R_2_ = 10 kΩ, the required R_b_ values are 50 kΩ, 24 kΩ, 12 kΩ, and 6 kΩ for selected resonant frequency values of 31.83 kHz, 66.31 kHz, 132.62 kHz, and 265.26 kΩ, respectively. Figure 24, Figure 25 and Figure 26 show the behavior of the AD844-based voltage-mode UAF resonant frequency f_o_ independently controlled by R_b_ when V_i3_ = V_in_ and V_i1_ = V_i2_ = 0. As shown in Figure 23 and Figure 26, simulations and measurements confirm the theoretical analysis according to (14) and (15).

### 3.2. The On-Chip CMOS VM-UAF Simulation and Measurement Results

Passive components of the on-chip CMOS CFOA-based voltage-mode UAF are designed as C_1_ = C_2_ = 15 pF and R_i_ = 20 kΩ (i = 1 to 6) with a resonance frequency of f_o_ = 530.5 kHz. Figure 27 shows the overall layout of the CFOA-based voltage-mode UAF and its chip micrograph with two CFOA-based voltage-mode UAFs. The CMOS implementation of CFOA is shown in Figure 28 [40]. In Figure 28, the length (L) and width (W) of transistors M1 to M16 are 0.4 μm and 75 μm, the L and W of transistors M17 to M20 are 0.8 μm and 13 μm, and the L and W of transistors M21 to M28 are 0.4 μm and 26 μm. Figure 29 shows the simulated frequency spectrum of the IBP filtering response at V_o1_. As shown in Figure 29, the THD is calculated as 0.25% for a sinusoidal input voltage of 0.4 V_pp_. Figure 30 shows the measured frequency spectrum of the IBP filtering response at V_o1_. As shown in Figure 30, the THD is calculated as 1% for a sinusoidal input voltage of 0.4 V_pp_, and the measured spurious-free dynamic range is 43.55 dBc. Figure 31, Figure 32 and Figure 33 also show the measurements for the CFOA-based voltage-mode UAF chip. The experimental and simulation results of the CFOA-based voltage-mode UAF chip relative to the theoretical analysis are shown in Figure 34, Figure 35 and Figure 36, respectively.

## 4. Conclusions

Three voltage-mode UAFs based on CFOA have been proposed in the technical literature, using four CFOAs, two grounded capacitors, five/six resistors, and one/two switches [43]. This study proposes a new voltage-mode UAF to improve the convenience and versatility of the recently introduced three voltage-mode UAF circuits [43]. The proposed voltage-mode UAF has three high-input and low-output impedances and can simultaneously realize LP, BP, HP, and BR filtering functions from the same configuration. Based on three CFOAs, the new voltage-mode UAF can be used in HP, LP, BP, BR, LPN, HPN, and AP without switches, providing the versatility and utility that active filter designers have expected. The proposed voltage-mode UAF has the following advantages: (1) configuration requires no switches and uses only three CFOAs, two grounded capacitors, and six resistors, (2) using two grounded capacitors is suitable for IC implementation, (3) the voltage-mode second-order LP, BP, HP, BR, LPN, HPN, and AP filtering functions are implemented from the same circuit configuration, (4) three high-input and low-output impedances are available for voltage-mode operation without needing voltage buffers, (5) the voltage-mode second-order LP, BP, HP, and BR second-order filtering functions are realized simultaneously without component matching conditions, (6) there is no series capacitor at the X-terminal of the CFOA, (7) the AP filtering function can be realized without voltage inverters or switches, (8) the ω_o_ and Q have independent controllability under certain circumstances, (9) the voltage-mode UAF has low active and passive sensitivity performance, and (10) The voltage-mode UAF circuit is implemented into a single CFOA-based chip. The CFOA-based voltage-mode UAF chip has lower power consumption and higher integration than the AD844-based filter. Experimental results from the commercially available off-the-shelf AD844 ICs and on-chip design measurements validate the theoretical analysis.

## Figures and Tables

**Figure 1 sensors-23-08258-f001:**
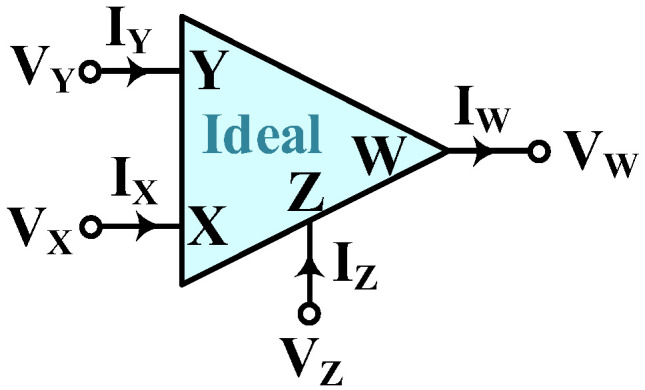
The circuit symbol of an ideal CFOA.

**Figure 2 sensors-23-08258-f002:**
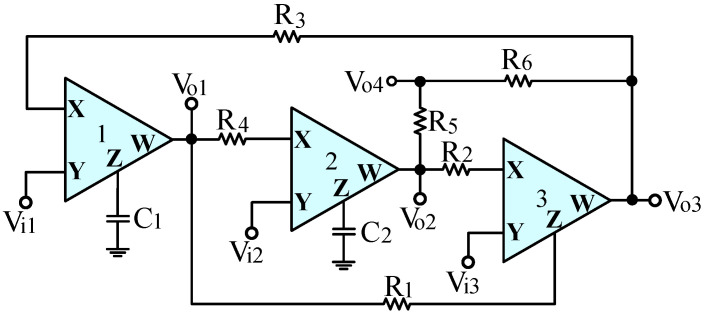
The proposed voltage-mode UAF configuration.

**Figure 3 sensors-23-08258-f003:**
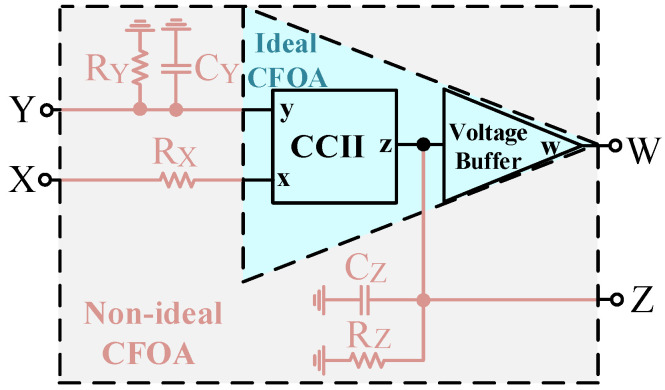
The effect of parasitic resistances and capacitances on ideal CFOA.

**Figure 4 sensors-23-08258-f004:**
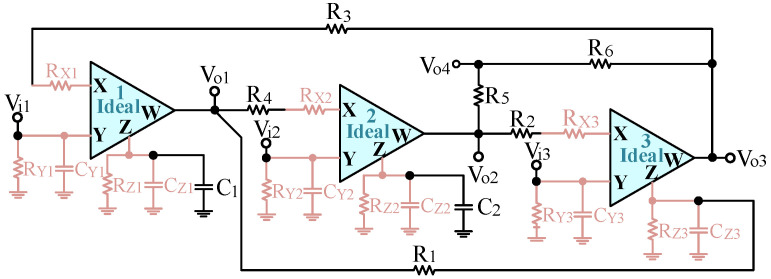
The effect of CFOA parasitic resistances and capacitances on the voltage-mode UAF configuration.

**Figure 5 sensors-23-08258-f005:**
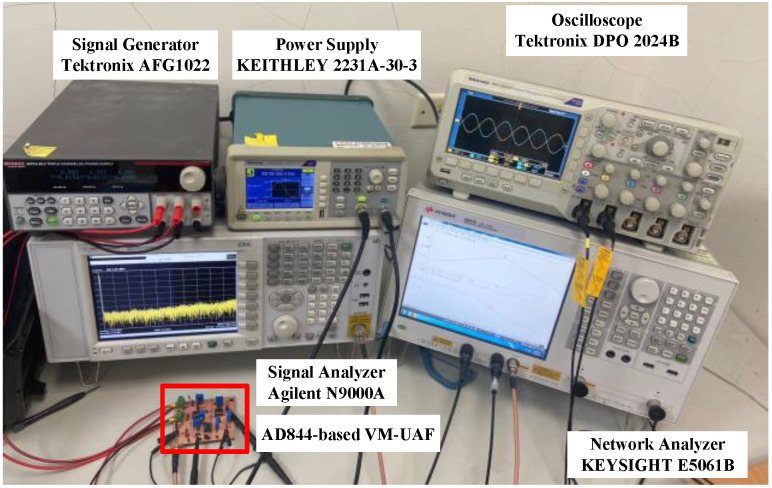
AD844-based voltage-mode UAF experimental setup platform.

**Figure 6 sensors-23-08258-f006:**
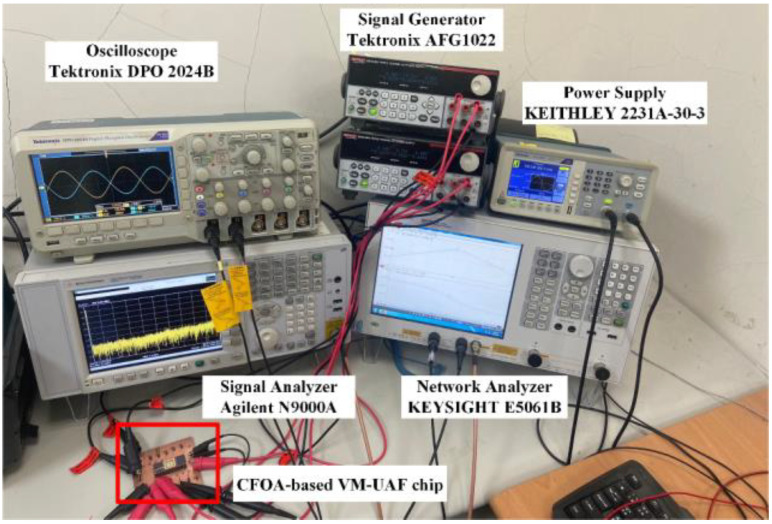
On-chip CMOS CFOA-based voltage-mode UAF experimental setup platform.

**Figure 7 sensors-23-08258-f007:**
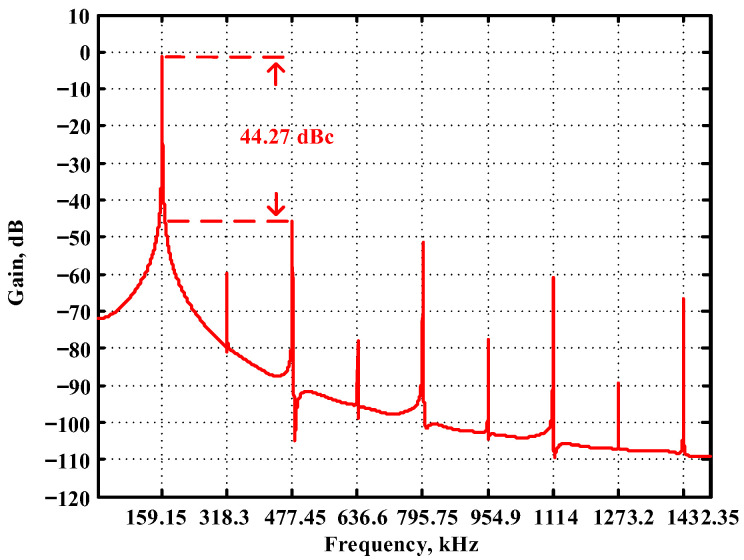
The simulated frequency spectrum for the AD844-based voltage-mode UAF at V_o1_ IBP filter.

**Figure 8 sensors-23-08258-f008:**
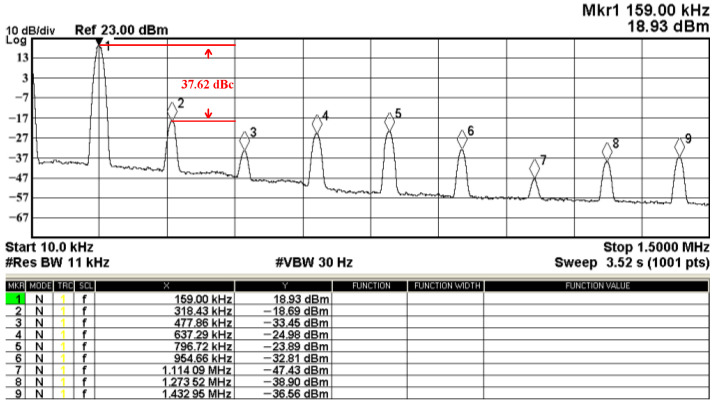
The measured frequency spectrum for the AD844-based voltage-mode UAF at V_o1_ IBP filter, where # is the reference symbol.

**Figure 9 sensors-23-08258-f009:**
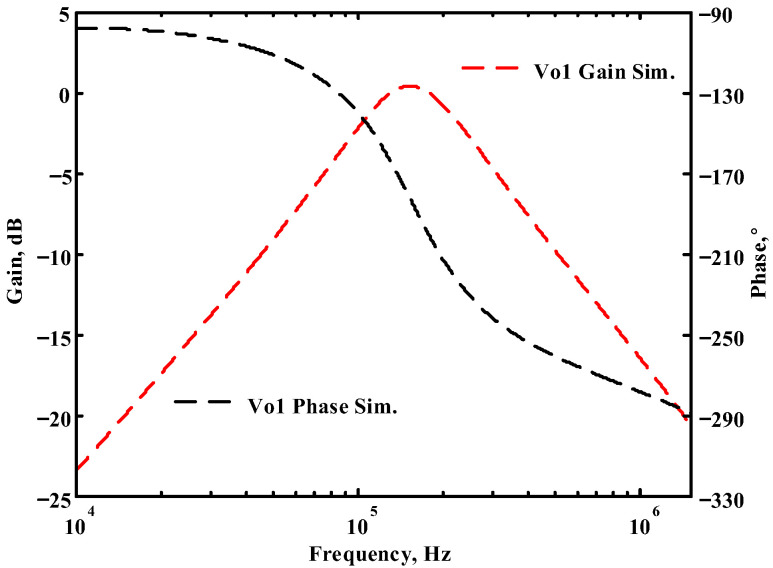
Simulation results for the AD844-based voltage-mode UAF at V_o1_ IBP filter.

**Figure 10 sensors-23-08258-f010:**
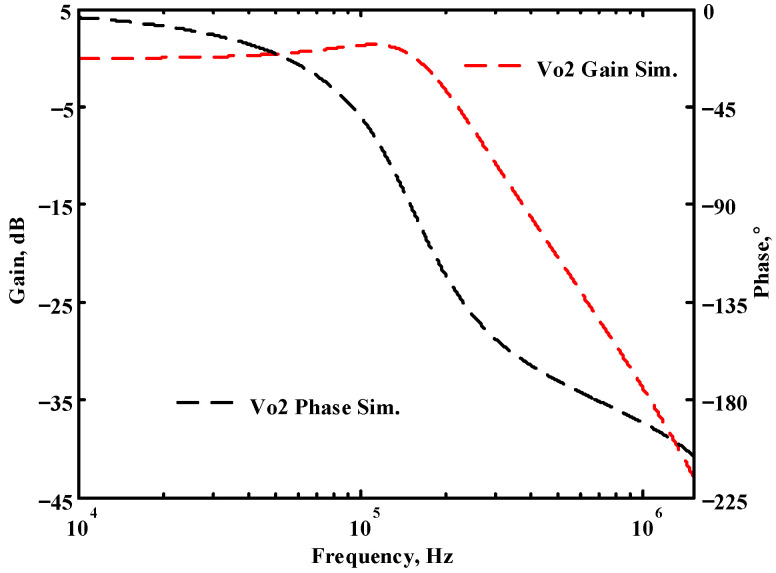
Simulation results for the AD844-based voltage-mode UAF at V_o2_ NLP filter.

**Figure 11 sensors-23-08258-f011:**
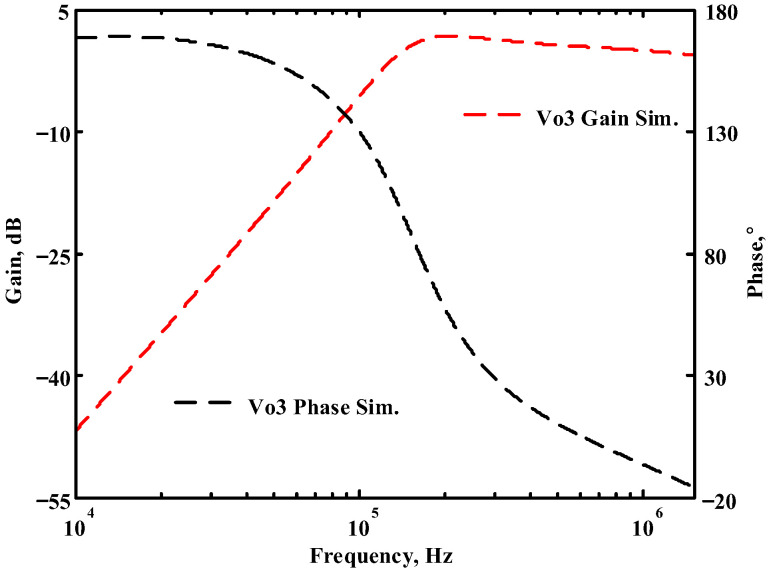
Simulation results for the AD844-based voltage-mode UAF at V_o3_ NHP filter.

**Figure 12 sensors-23-08258-f012:**
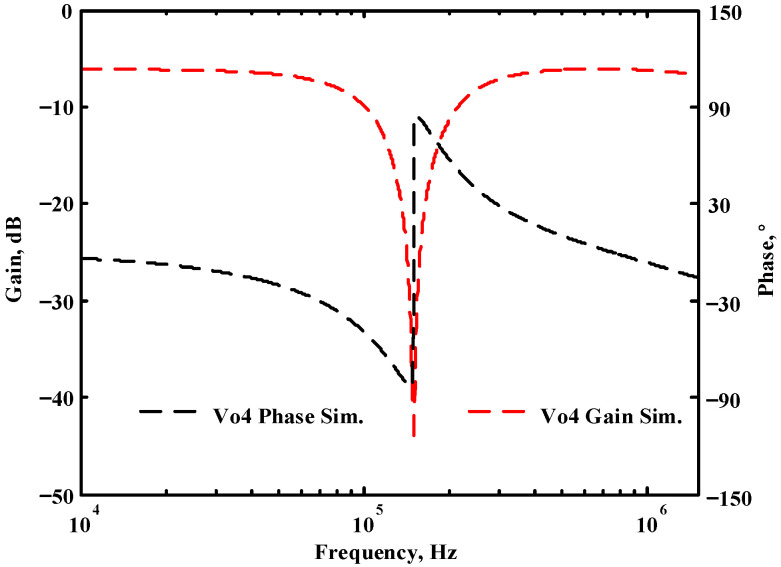
Simulation results for the AD844-based voltage-mode UAF at V_o4_ BR filter.

**Figure 13 sensors-23-08258-f013:**
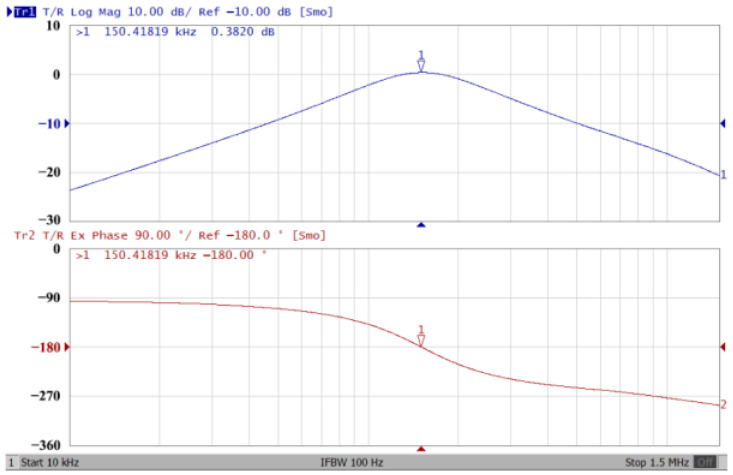
Gain (**top**) and phase (**bottom**) measurement results for the AD844-based voltage-mode UAF at V_o1_ IBP filter.

**Figure 14 sensors-23-08258-f014:**
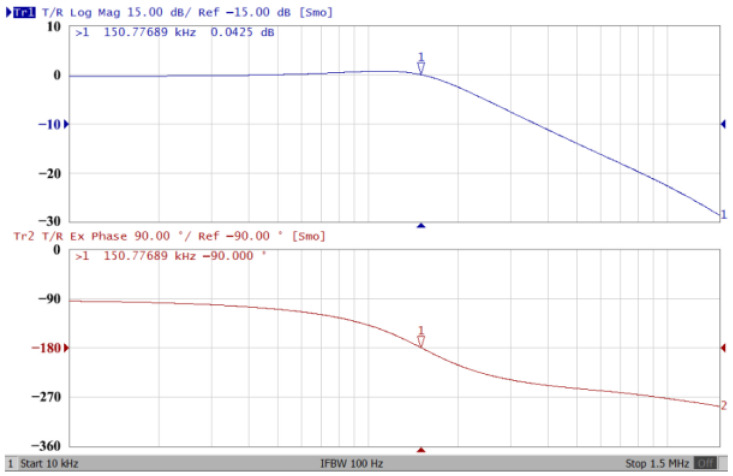
Gain (**top**) and phase (**bottom**) measurement results for the AD844-based voltage-mode UAF at V_o2_ NLP filter.

**Figure 15 sensors-23-08258-f015:**
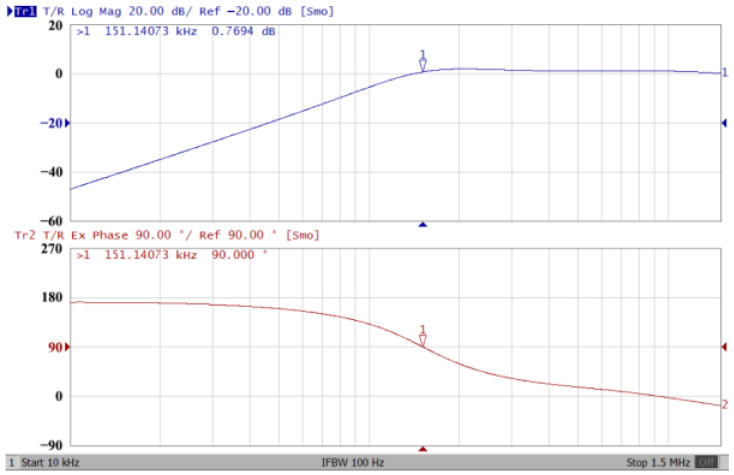
Gain (**top**) and phase (**bottom**) measurement results for the AD844-based voltage-mode UAF at V_o3_ NHP filter.

**Figure 16 sensors-23-08258-f016:**
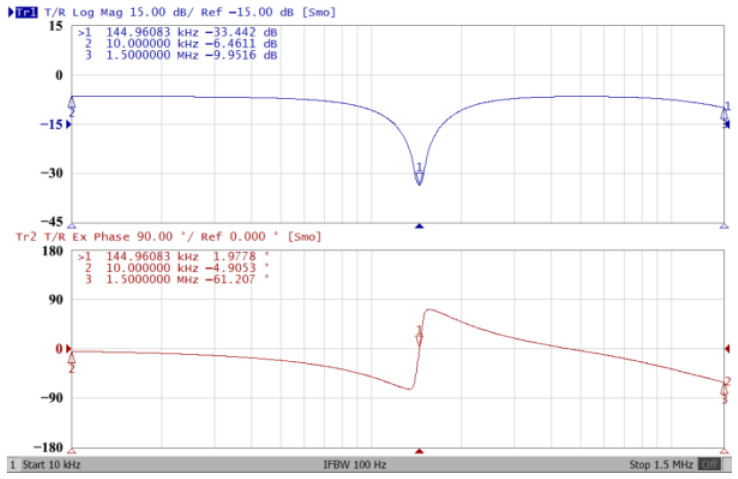
Gain (**top**) and phase (**bottom**) measurement results for the AD844-based voltage-mode UAF at V_o4_ BR filter.

**Figure 17 sensors-23-08258-f017:**
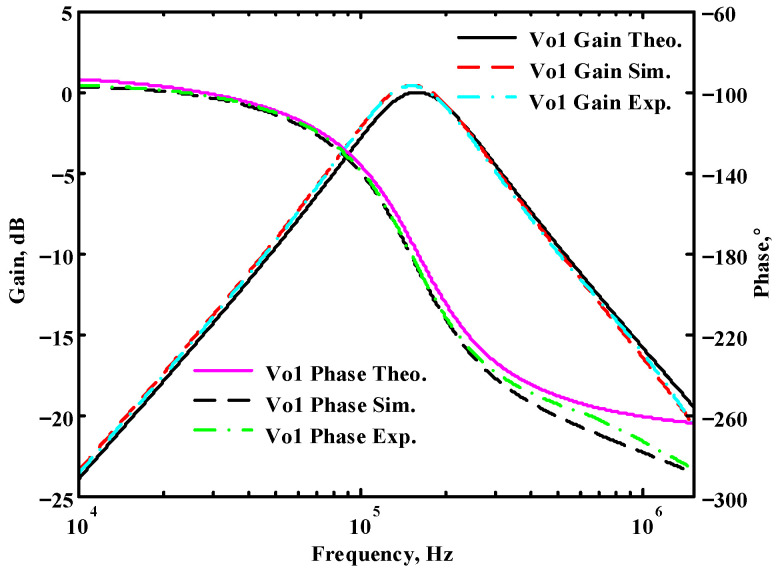
Ideal, simulated, and measured results for the AD844-based voltage-mode UAF at V_o1_ IBP filter.

**Figure 18 sensors-23-08258-f018:**
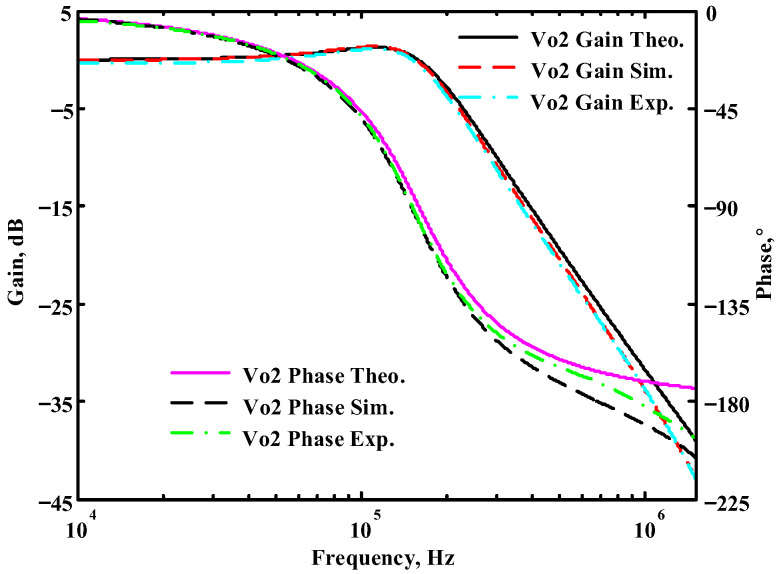
Ideal, simulated, and measured results for the AD844-based voltage-mode UAF at V_o2_ NLP filter.

**Figure 19 sensors-23-08258-f019:**
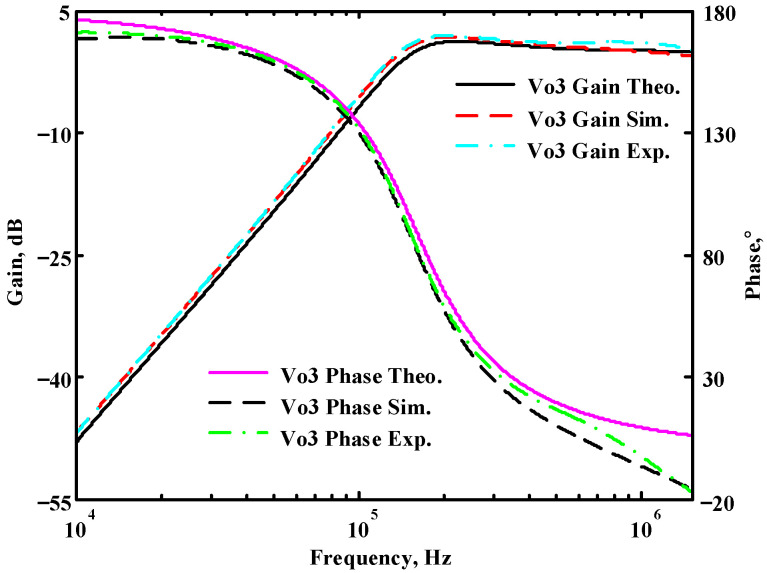
Ideal, simulated, and measured results for the AD844-based voltage-mode UAF at V_o3_ NHP filter.

**Figure 20 sensors-23-08258-f020:**
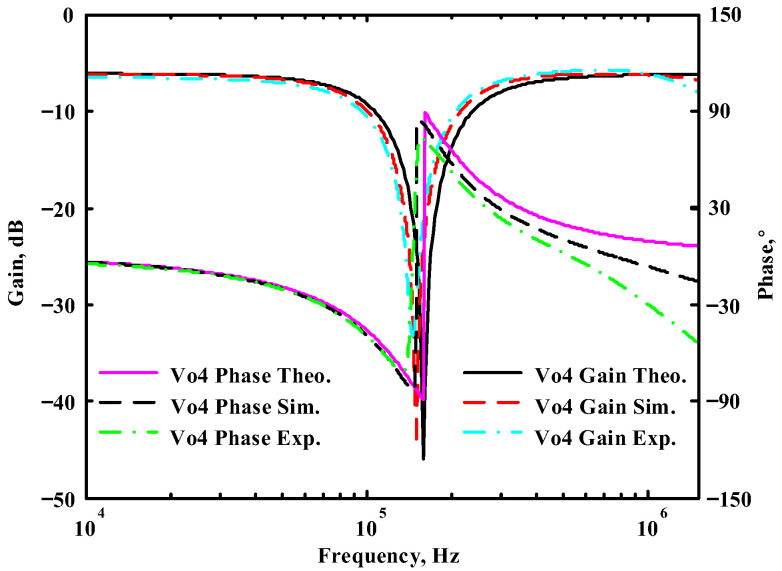
Ideal, simulated, and measured results for the AD844-based voltage-mode UAF at V_o4_ BR filter.

**Figure 21 sensors-23-08258-f021:**
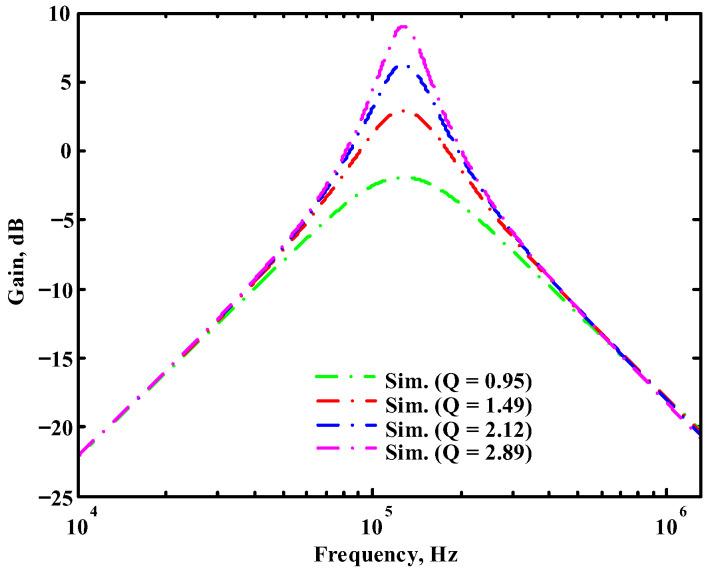
The simulated quality factor for the AD844-based voltage-mode UAF.

**Figure 22 sensors-23-08258-f022:**
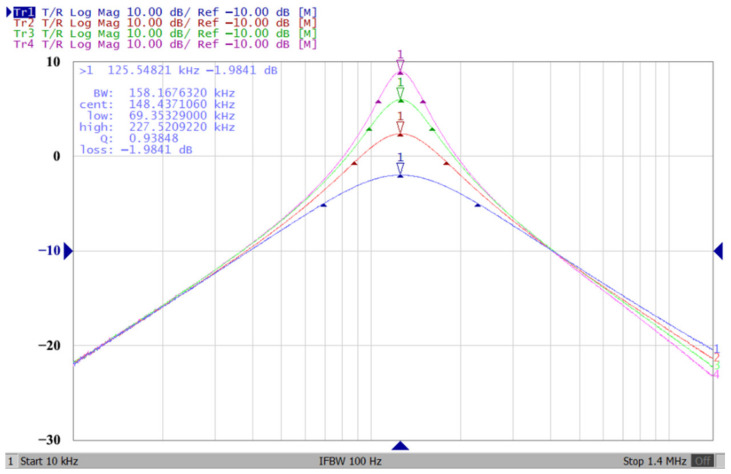
The measured quality factor for the AD844-based voltage-mode UAF.

**Figure 23 sensors-23-08258-f023:**
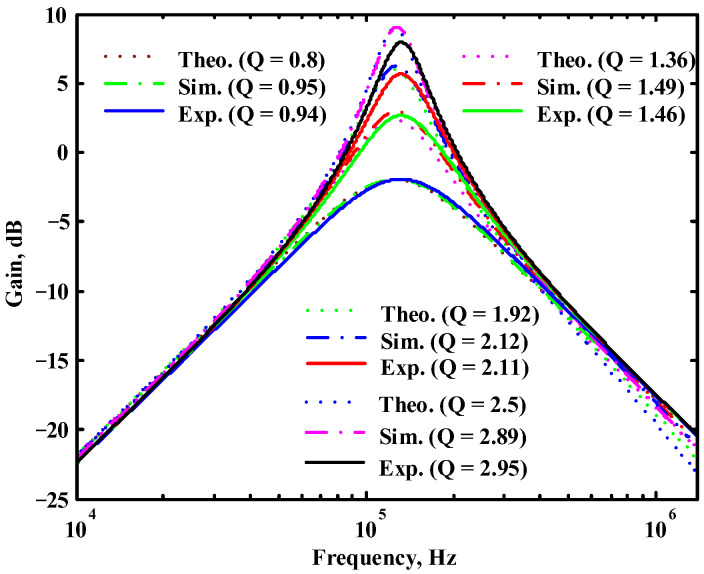
Ideal, simulated, and measured results for independent quality factor control.

**Figure 24 sensors-23-08258-f024:**
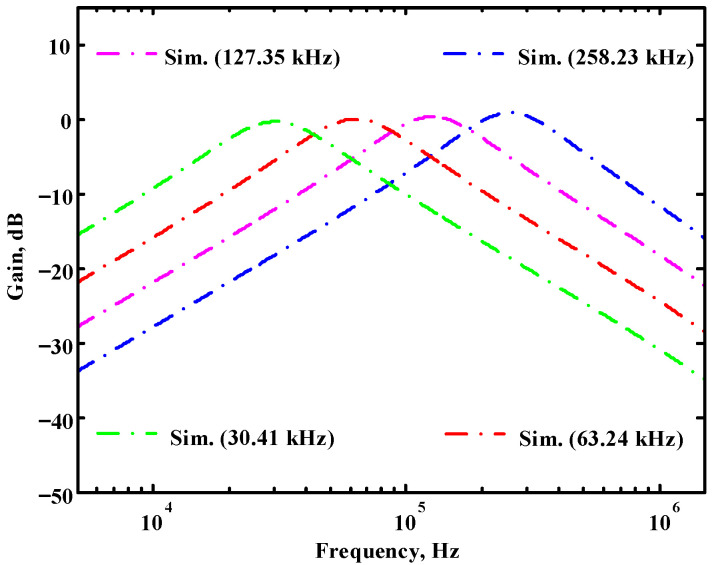
The simulated resonant frequency for the AD844-based voltage-mode UAF without affecting the parameter Q.

**Figure 25 sensors-23-08258-f025:**
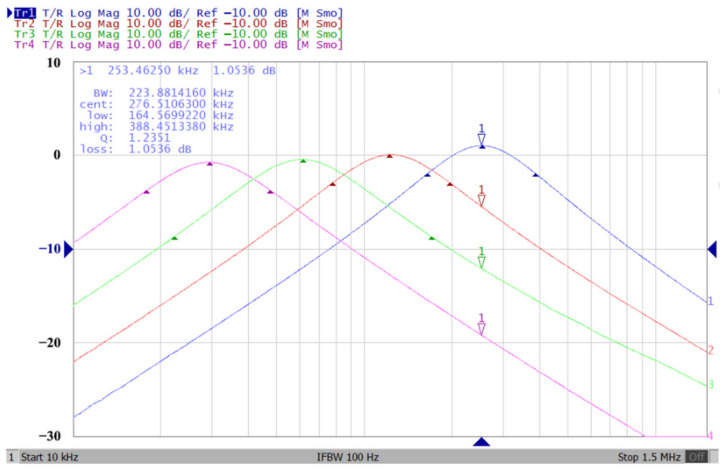
The measured resonant frequency for the AD844-based voltage-mode UAF without affecting the parameter Q.

**Figure 26 sensors-23-08258-f026:**
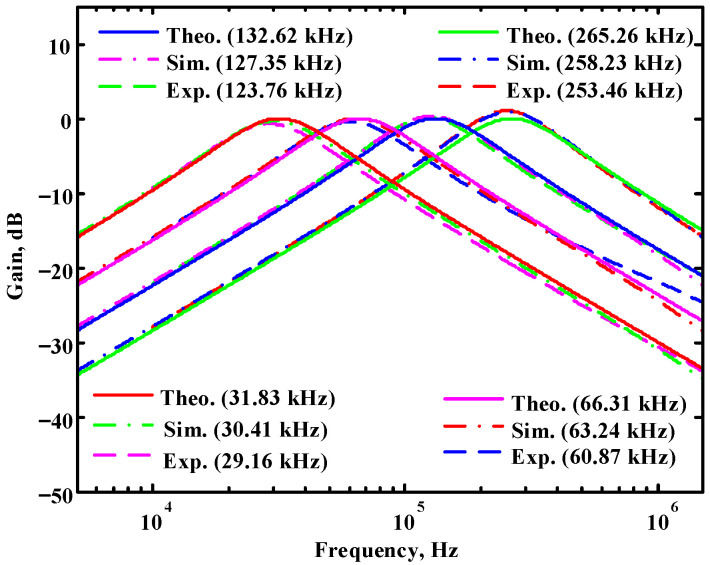
Ideal, simulated, and measured results for independent resonant frequency control without affecting the parameter Q.

**Figure 27 sensors-23-08258-f027:**
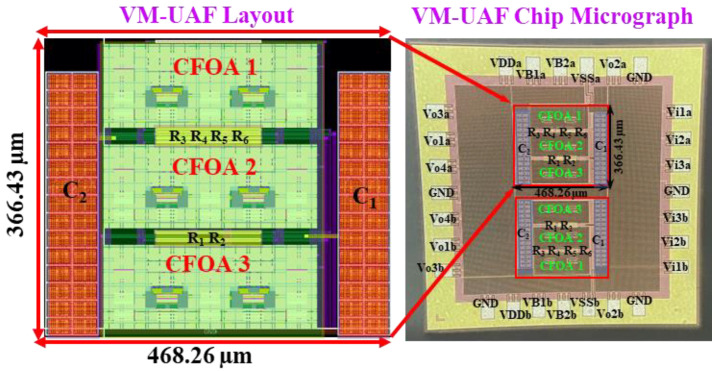
The overall layout of the CFOA-based voltage-mode UAF and its micrograph of a chip with two CFOA-based voltage-mode UAFs.

**Figure 28 sensors-23-08258-f028:**
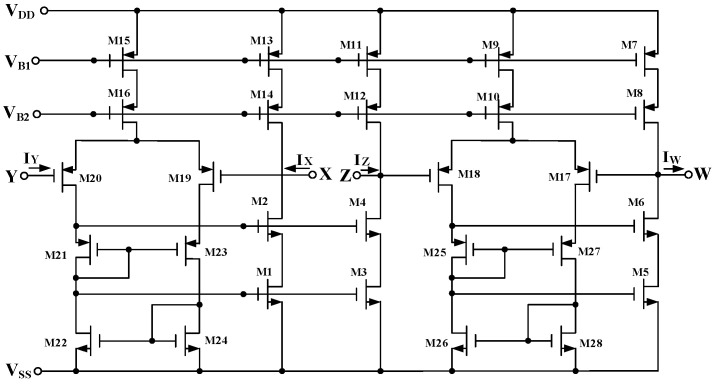
The CMOS implementation of CFOA.

**Figure 29 sensors-23-08258-f029:**
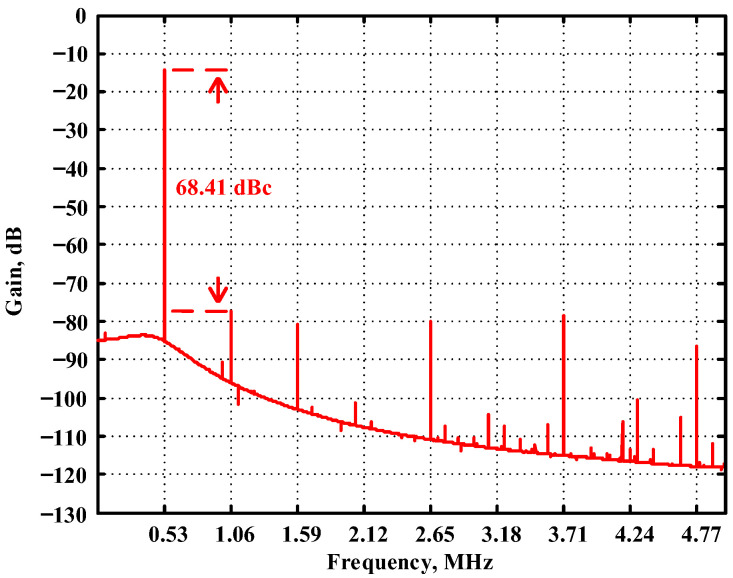
The simulated frequency spectrum for the CFOA-based voltage-mode UAF chip at V_o1_ IBP filter.

**Figure 30 sensors-23-08258-f030:**
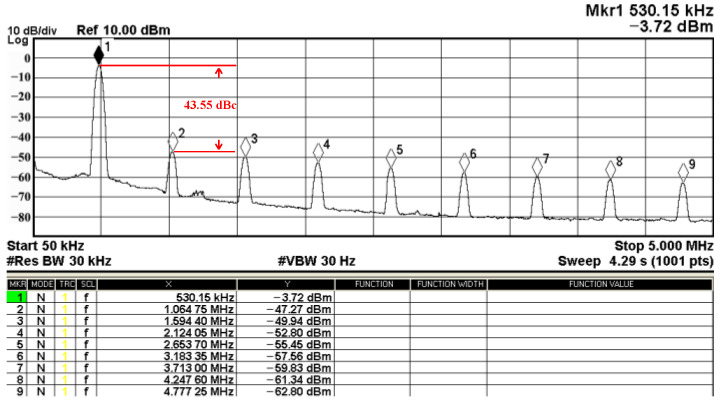
The measured frequency spectrum for the CFOA-based voltage-mode UAF chip at V_o1_ IBP filter, where # is the reference symbol.

**Figure 31 sensors-23-08258-f031:**
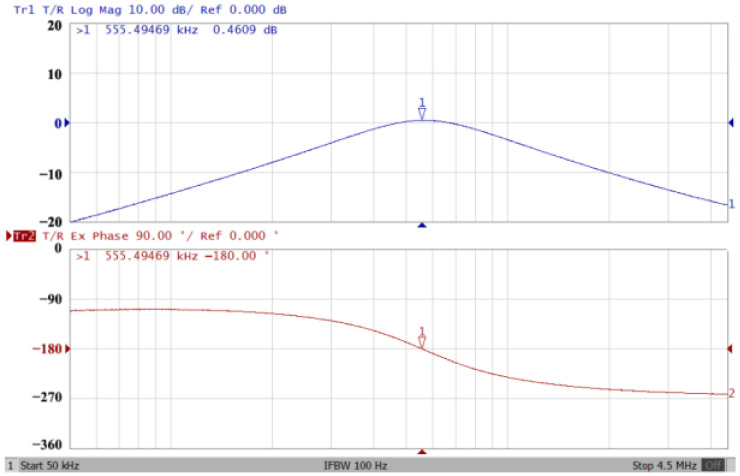
Gain (**top**) and phase (**bottom**) simulation results for the CFOA-based voltage-mode UAF chip at V_o1_ IBP filter.

**Figure 32 sensors-23-08258-f032:**
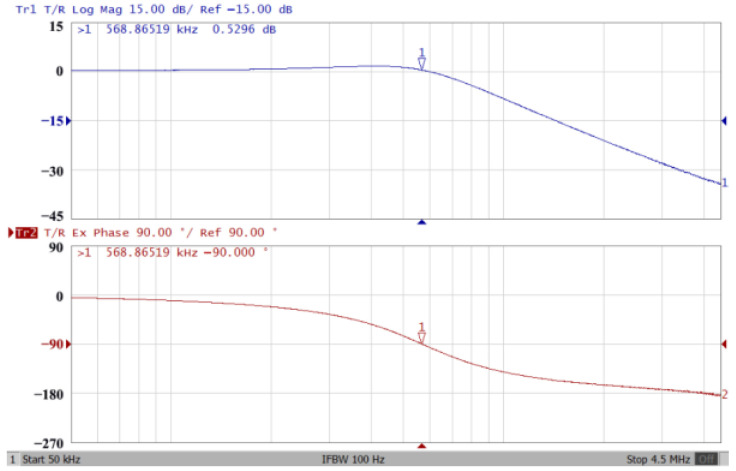
Gain (**top**) and phase (**bottom**) simulation results for the CFOA-based voltage-mode UAF chip at V_o2_ NLP filter.

**Figure 33 sensors-23-08258-f033:**
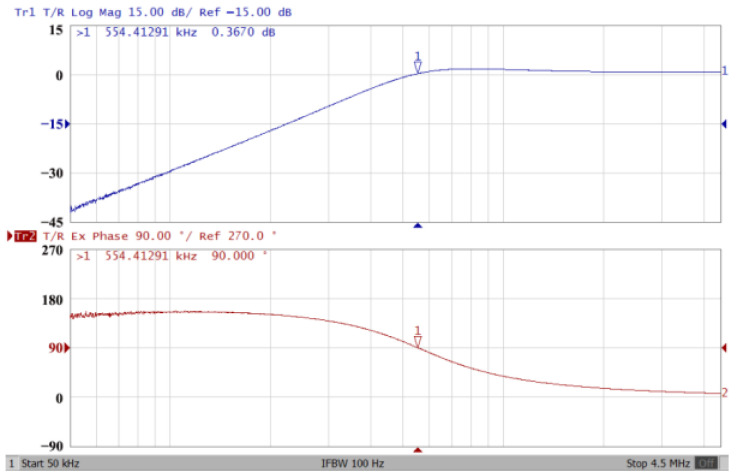
Gain (**top**) and phase (**bottom**) simulation results for the CFOA-based voltage-mode UAF chip at V_o3_ NHP filter.

**Figure 34 sensors-23-08258-f034:**
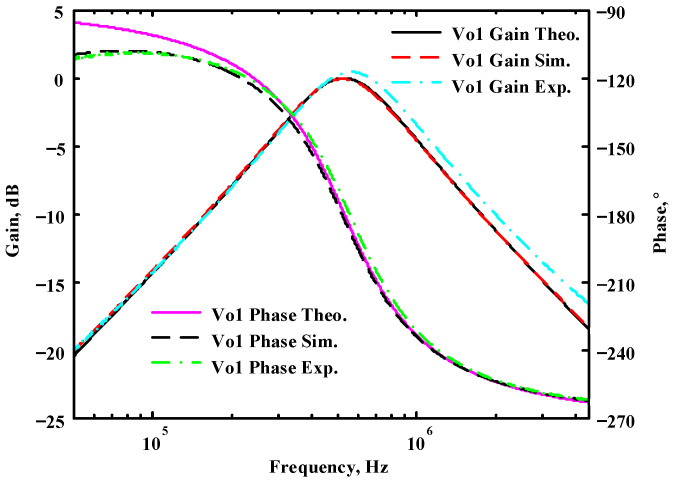
Ideal, simulated, and measured results for the CFOA-based voltage-mode UAF chip at V_o1_ IBP filter.

**Figure 35 sensors-23-08258-f035:**
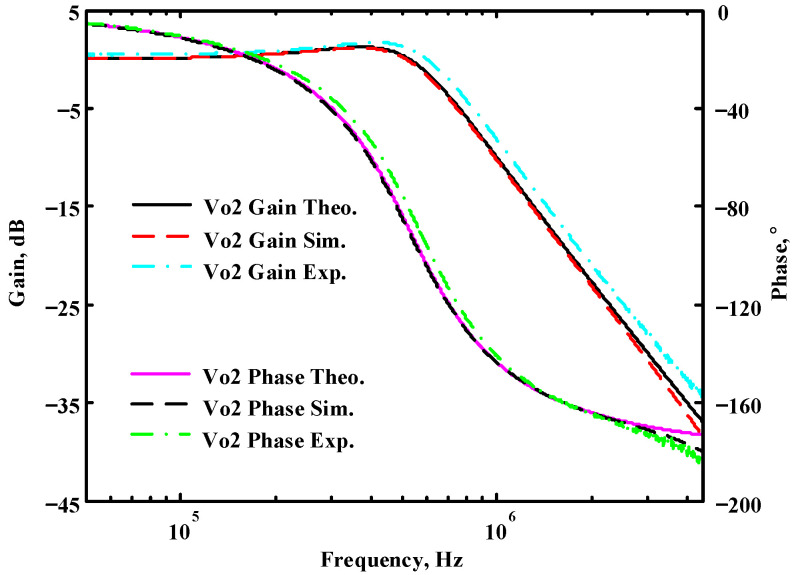
Ideal, simulated, and measured results for the CFOA-based voltage-mode UAF chip at V_o2_ NLP filter.

**Figure 36 sensors-23-08258-f036:**
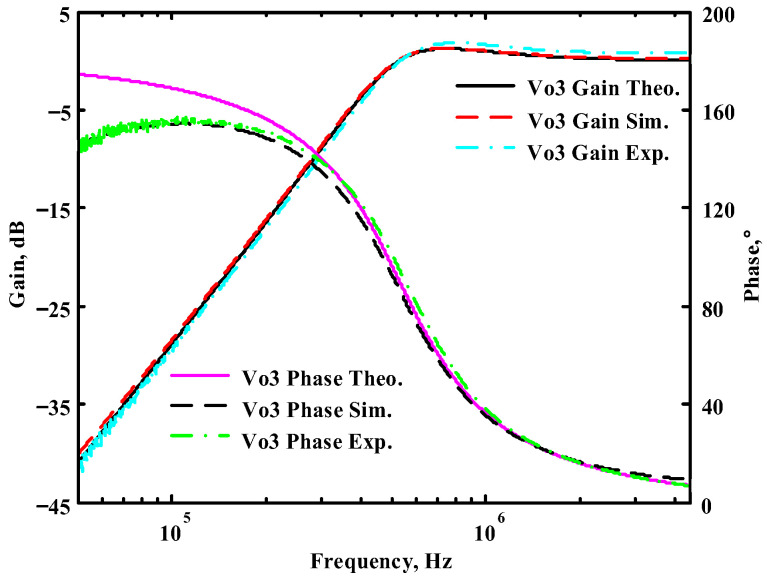
Ideal, simulated, and measured results for the CFOA-based voltage-mode UAF chip at V_o3_ NHP filter.

**Table 1 sensors-23-08258-t001:** Comparison of the proposed CFOA-based voltage-mode UAF with previously published filters.

Topology/Year	Number of Active and Passive Elements	Number of Filtering Functions	Simultaneous Realization of Filtering Functions	Use of Only Grounded Capacitors	No Series Capacitor on the X Terminal of the CFOA	No Switches Required	Orthogonal Control of Q and ω_o_	Operating Resonance Frequency f_o_	Measured Spurious-Free Dynamic Range	Measured Total Harmonic Distortion	Supply and Power	Results/ Technology
[32]/2005	CFOA × 4 R × 6 C × 2	3 (LP, BP, HP)	LP, BP, HP	Yes	Yes	Yes	Yes	15.9 kHz	NA	NA	NA	Sim./AD844
[33]/2005	CFOA × 3 R × 3 C × 2	5 (LP, BP, HP, BR, AP)	BP, HP, BR	No	Yes	Yes	Yes	15.92 kHz	NA	NA	NA	Sim./AD844
[34]/2005	CFOA × 4 R × 4 C × 2	4 (LP, BP, HP, BR)	LP, BP, HP, BR	Yes	Yes	Yes	Yes	7.96 kHz	NA	NA	NA	Exp./AD844
[35]/2005	CFOA × 3 R × 4 C × 2	3 (LP, BP, HP)	LP, BP, HP	Yes	No	Yes	No	5.68 kHz	NA	NA	±12 V	Sim./AD844
[36] in Figure 1d/2006	CFOA × 4 R × 8 C × 2 Switch × 1	5 (LP, BP, HP, BR, AP)	LP, BP, HP, BR/AP	Yes	Yes	No	No	5.62 kHz	NA	NA	±12 V	Exp./AD844
[37]/2019	CFOA × 3 R × 3 C × 2	4 (LP, BP, HP, BR)	LP, BP, BR	Yes	Yes	Yes	Yes	39.79 kHz	NA	0.56% @ 2 V_pp_	±6 V, 180 mW	Exp./AD844
[38]/2020	CFOA × 3 R × 4 C × 2	3 (LP, BP, BR)	LP, BP, BR	Yes	Yes	Yes	Yes	102 kHz	NA	NA	±6 V, 168 mW	Exp./AD844
[39]/2021	CFOA × 3 R × 3 C × 2	3 (LP, BP, BR)	LP, BP, BR	Yes	Yes	Yes	Yes	39.79 kHz	NA	NA	±6 V, 255 mW	Exp./AD844
[40]/2021	CFOA × 3 R × 4 C × 2	4 (LP, BP, HP, BR)	LP, BP, HP	Yes	Yes	Yes	Yes	117.9 kHz (for the AD844) 757.88 kHz (for the chip)	NA	3.2% @ 5.68 V_pp_ (for the AD844) 3.18% @ 1.2 V_pp_ (for the chip)	±6 V, 168 mW (for the AD844) ±0.9 V, 5.4 mW (for the chip)	Exp./AD844 & Chip (TSMC 180 nm)
[41]/2022	CFOA × 3 R × 4 C × 2	3 (LP, BP, BR)	LP, BP, BR	Yes	Yes	Yes	Yes	568 kHz	31.63 dBc	3.3% @ 1.2 V_pp_	±0.9 V, 5.4 mW	Exp./Chip (TSMC 180 nm)
[42]/2023	CFOA × 3 R × 5 C × 2	3 (LP, BP, BR)	LP, BP, BR	Yes	Yes	Yes	Yes	530 kHz	33.74 dBc	2.88% @ 1.5 V_pp_	±0.9 V, 5.4 mW	Exp./Chip (TSMC 180 nm)
[43] in Figure 1a/2022	CFOA × 4 R × 5 C × 2 Switch × 2	5 (LP, BP, HP, BR, AP)	LP, BP, BR, HP/AP	Yes	Yes	No	Yes	159 kHz	NA	NA	±10 V	Exp./AD844
[43] in Figure 1b/2022	CFOA × 4 R × 6 C × 2 Switch × 2	5 (LP, BP, HP, BR, AP)	LP, BP, BR, HP/AP	Yes	Yes	No	Yes	159 kHz	NA	NA	±10 V	Exp./AD844
[43] in Figure 1 c/2022	CFOA × 4 R × 6 C × 2 Switch × 1	5 (LP, BP, HP, BR, AP)	LP, BP, BR, HP/AP	Yes	Yes	No	Yes	159 kHz	NA	NA	±10 V	Exp./AD844
Proposed	CFOA × 3 R × 6 C × 2	7 (LP, BP, HP, BR, LPN, HPN, AP)	LP, BP, HP, BR/ LPN/ HPN	Yes	Yes	Yes	Yes	159.15 kHz (for the AD844) 530.5 kHz (for the chip)	37.62 dBc (for the AD844) 43.55 dBc (for the chip)	1.68% @ 5.2 V_pp_ (for the AD844) 1% @ 0.4 V_pp_ (for the chip)	±6 V, 168 mW (for the AD844) ±0.9 V, 3.6 mW (for the chip)	Exp./AD844 & Chip (TSMC 180 nm)

## Data Availability

Not applicable.

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
