# Peer review of "Design and Verification of a New Universal Active Filter Based on the Current Feedback Operational Amplifier and Commercial AD844 Integrated Circuit"

_sensors, 2023, doi:10.3390/s23198258_

Round 1

Reviewer 1 Report

This paper presents a triple-input-four-output (TIFO)-type voltage-mode (VM) universal active filter (VM-UAF) based on three current-feedback operational amplifiers (CFOAs). I have some questions, as following:
1. How to adjust the bandwidth of the proposed filter.

2. The selectivity should be compared.

3. The power dissipation should be given.

4. Some of the picture headers need to be refined.

None

Author Response

The authors would like to thank anonymous reviewers for carefully reviewing the paper and for many thoughtful comments on the original manuscript. The manuscript has been revised and improved according to the suggestions of reviewers. The changes in the revised manuscript are marked in blue.

Reply to Reviewer 1 comments

Comments and Suggestions for Authors

This paper presents a triple-input-four-output (TIFO)-type voltage-mode (VM) universal active filter (VM-UAF) based on three current-feedback operational amplifiers (CFOAs). I have some questions, as following:

  1. How to adjust the bandwidth of the proposed filter.

Ans: Thanks for your comment. According to (9), the voltage-mode UAF parameters bandwidth (BW), ωo, and Q can be controlled orthogonally by R1 = R3 = Ra. In this case, Equation (9) becomes (14), and Ra independently controls the parameters BW and Q without affecting the parameter ωo. Please see page 8, from lines 160 to 163, of the revised manuscript.

  1. The selectivity should be compared.

Ans: Thanks for your comment. Table 1 has been modified, and the critical performance has text descriptions. Please see page 3, from lines 111 to 123, of the revised manuscript.

  1. The power dissipation should be given.

Ans: Thanks for your comment. The AD844-based VM-UAF has a supply voltage of 12 V (±6 V) and a power consumption of 168 mW. The on-chip CMOS CFOA-based VM-UAF has a supply voltage of 1.8 V (±0.9 V) and a power consumption of 3.6 mW. Please see page 12, from lines 217 to 219, of the revised manuscript.

  1. Some of the picture headers need to be refined.

Ans: Thanks for your comment. We have improved the figure headers. The changes in the revised manuscript are marked in blue.

Reviewer 2 Report

The authors of the paper “Design and Verification of a New Universal Active Filter Based on the Current Feedback Operational Amplifier and Commercial AD844 Integrated Circuit” presents a widely configurable CFOAs-based topology. The presented topology is simulated and implemented with two technologies. The paper is well-written and documented but not sufficiently well elaborated from a graphical or layout perspective. From a style perspective I think that defining possible uses in sensor applications should be a central element. This justification, exemplification would bring a plus to the paper and a substantial motivation of the proposed topology.

1. The paper abuses (many atypical) notations already in the abstract. This style is not beneficial for readers who have to swim through a multitude of acronyms (example - AD844-based filter using three commercially available off-the-shelf AD844s, 2GCs, and 6Rs).

2. Without row numbering it is difficult to indicate too many particular observations, or as many as necessary.

3. As far as possible do not use grammatical structures specific to a marketing text. I invite you to remain in the academic area with specific modesty. Phrase - To the authors' best knowledge, any VM-UAF employing only three CFOAs ... is already in the literature an unforgivable plague (playing dumb by playing stupid).

4. Many sentences repeat themselves either aditeren or by content. I invite you to review the manuscript carefully.

5. I recommend that you retain only the essential figures that provide clear evidence of the electrical performance of the proposed topology. At the moment there are many figures that do not make a real contribution to the demonstration of performance. Find a filter and I don't think that the response in the time domain proves anything other than that the circuit works.

6. A compression of the paper to the elements that relate to the electrical performance of the proposed topology and its obligatory usefulness in sensing would add value to this article.

Author Response

The authors would like to thank anonymous reviewers for carefully reviewing the paper and for many thoughtful comments on the original manuscript. The manuscript has been revised and improved according to the suggestions of reviewers. The changes in the revised manuscript are marked in blue.

Reply to Reviewer 2 comments

The authors of the paper “Design and Verification of a New Universal Active Filter Based on the Current Feedback Operational Amplifier and Commercial AD844 Integrated Circuit” presents a widely configurable CFOAs-based topology. The presented topology is simulated and implemented with two technologies. The paper is well-written and documented but not sufficiently well elaborated from a graphical or layout perspective. From a style perspective I think that defining possible uses in sensor applications should be a central element. This justification, exemplification would bring a plus to the paper and a substantial motivation of the proposed topology.

Ans: Many thanks for the reviewer’s positive feedback. The manuscript has been revised and improved according to the reviewer’s suggestion. The changes in the revised manuscript are marked in blue.

  1. The paper abuses (many atypical) notations already in the abstract. This style is not beneficial for readers who have to swim through a multitude of acronyms (example - AD844-based filter using three commercially available off-the-shelf AD844s, 2GCs, and 6Rs).

Ans: Thanks for your comment. We have improved the abbreviation notation in the abstract. Please see the abstract of the revised manuscript on page 1.

  1. Without row numbering it is difficult to indicate too many particular observations, or as many as necessary.

Ans: Thanks for your comment. We have added the line numbers of the revised manuscript.

  1. As far as possible do not use grammatical structures specific to a marketing text. I invite you to remain in the academic area with specific modesty. Phrase - To the authors' best knowledge, any VM-UAF employing only three CFOAs ... is already in the literature an unforgivable plague (playing dumb by playing stupid).

Ans: Thanks for your comment. We improved the sentences described. Please see page 3, from lines 111 to 123, of the revised manuscript.

  1. Many sentences repeat themselves either aditeren or by content. I invite you to review the manuscript carefully.

Ans: Thanks for your comment. We improved the sentences described. The changes in the revised manuscript are marked in blue.

  1. I recommend that you retain only the essential figures that provide clear evidence of the electrical performance of the proposed topology. At the moment there are many figures that do not make a real contribution to the demonstration of performance. Find a filter and I don't think that the response in the time domain proves anything other than that the circuit works.

Ans: Thanks for your comment. We have removed the time domain simulations and measurements from the original manuscript. Please see the Section 3 simulation and experimental results from page 12 of the revised manuscript.

  1. A compression of the paper to the elements that relate to the electrical performance of the proposed topology and its obligatory usefulness in sensing would add value to this article.

Ans: Thanks for your comment. We describe this in the introduction of the revised manuscript. Please see page 1, Section 1 Introduction, lines 32 to 38 of the revised manuscript.

Reviewer 3 Report

This work presents a three-input, four-output universal active filter that operates in voltage mode. The filter uses current feedback operational amplifiers (CFOA), resistors, and grounded capacitors to achieve its functionality. The filter has three high-impedance inputs and three low-impedance outputs, which allow for cascading without voltage buffers. The filter can perform low-pass, high-pass, band-reject, band-pass, and all-pass filtering functions without using voltage inverters or switches. Additionally, the filter can perform two unique filtering functions, low-pass notch and high-pass notch.

To validate the proposed solution, the authors implemented the filter using three AD844s, two grounded capacitors, and six resistors. They then tested the circuit using OrCad PSpice simulation and experimental measurements. The authors also implemented the circuit into a chip using three CFOAs, two grounded capacitors, and six resistors. They then tested the circuit using HSpice for post-layout simulation experimental measurements.

The authors provide an overview of past research from the specialized literature in the first chapter. They present it in a table, but upon initial inspection, the cited articles seem outdated. This may lead one to believe that the topic is no longer relevant. Can you please supplement the bibliography with more recent articles?

The second chapter provides a thorough explanation of the active filter design proposed by the authors. It covers the key relationships and how the primary resistances and parasitic capacitances of the CFBOAs affect the filter's performance.  It is worth noting that rewriting relations 26-28, and 31-33 as 1-3 would be beneficial.

The third chapter presents a comparison between theoretical results, simulations, and experimental measurements. During the simulation of the AD844 circuit in OrCad PSpice, which model was used? Also, were all the parameters used for sensitivity analysis included in the model?

Author Response

The authors would like to thank anonymous reviewers for carefully reviewing the paper and for many thoughtful comments on the original manuscript. The manuscript has been revised and improved according to the suggestions of reviewers. The changes in the revised manuscript are marked in blue.

Reply to Reviewer 3 comments

  1. This work presents a three-input, four-output universal active filter that operates in voltage mode. The filter uses current feedback operational amplifiers (CFOA), resistors, and grounded capacitors to achieve its functionality. The filter has three high-impedance inputs and three low-impedance outputs, which allow for cascading without voltage buffers. The filter can perform low-pass, high-pass, band-reject, band-pass, and all-pass filtering functions without using voltage inverters or switches. Additionally, the filter can perform two unique filtering functions, low-pass notch and high-pass notch.

Ans: Many thanks for the reviewer’s positive feedback.

  1. To validate the proposed solution, the authors implemented the filter using three AD844s, two grounded capacitors, and six resistors. They then tested the circuit using OrCad PSpice simulation and experimental measurements. The authors also implemented the circuit into a chip using three CFOAs, two grounded capacitors, and six resistors. They then tested the circuit using HSpice for post-layout simulation experimental measurements.

Ans: Many thanks for the reviewer’s positive feedback.

  1. The authors provide an overview of past research from the specialized literature in the first chapter. They present it in a table, but upon initial inspection, the cited articles seem outdated. This may lead one to believe that the topic is no longer relevant. Can you please supplement the bibliography with more recent articles?

Ans: Thanks for your comment. Table 1 has been modified, and the critical performance has text descriptions. Please see page 3, from lines 111 to 123, of the revised manuscript.

  1. The second chapter provides a thorough explanation of the active filter design proposed by the authors. It covers the key relationships and how the primary resistances and parasitic capacitances of the CFBOAs affect the filter's performance. It is worth noting that rewriting relations 26-28, and 31-33 as 1-3 would be beneficial.

Ans: Thanks for your comment. In this case, the three non-ideal voltage transfer functions, the denominator Dn(s), and the non-ideal filter parameters of ωon and Qn were obtained as Equations (34) to (39). Please see page 11, from lines 205 to 207, of the revised manuscript.

  1. The third chapter presents a comparison between theoretical results, simulations, and experimental measurements. During the simulation of the AD844 circuit in OrCad PSpice, which model was used? Also, were all the parameters used for sensitivity analysis included in the model?

Ans: Thanks for your comment. Simulations were performed using the built-in library of AD844/AD model parameters of OrCAD PSpice software, and OrCAD PSpice software features sensitivity/Monte Carlo analysis capabilities. Please see page 12, from lines 223 to 225, of the revised manuscript.

Round 2

Reviewer 2 Report

The authors of the revised version of the paper “Design and Verification of a New Universal Active Filter Based on the Current Feedback Operational Amplifier and Commercial AD844 Integrated Circuit” have substantially completed this version with better graphic, tablet and explanatory content in line with the recommendations to the original version. The work in this version is better developed and structured in accordance with general practice.  In its present form, I consider that the paper meets the conditions for publication.